# Practical and Optimal LSH for Angular Distance

**Alexandr Andoni**[*]
Columbia University

**Piotr Indyk**
MIT

**Thijs Laarhoven**
TU Eindhoven

**Ilya Razenshteyn**
MIT

**Ludwig Schmidt**
MIT

## Abstract

We show the existence of a Locality-Sensitive Hashing (LSH) family for the angular distance that yields an approximate Near Neighbor Search algorithm with the asymptotically optimal running time exponent. Unlike earlier algorithms with this property (e.g., Spherical LSH [1, 2]), our algorithm is also practical, improving upon the well-studied hyperplane LSH [3] in practice. We also introduce a *multiprobe* version of this algorithm and conduct an experimental evaluation on real and synthetic data sets.

We complement the above positive results with a fine-grained lower bound for the quality of any LSH family for angular distance. Our lower bound implies that the above LSH family exhibits a trade-off between evaluation time and quality that is close to optimal for a natural class of LSH functions.

## 1 Introduction

Nearest neighbor search is a key algorithmic problem with applications in several fields including computer vision, information retrieval, and machine learning [4]. Given a set of $n$ points $P \subset \mathbb{R}^d$, the goal is to build a data structure that answers nearest neighbor queries efficiently: for a given query point $q \in \mathbb{R}^d$, find the point $p \in P$ that is closest to $q$ under an appropriately chosen distance metric. The main algorithmic design goals are usually a fast query time, a small memory footprint, and—in the approximate setting—a good quality of the returned solution.

There is a wide range of algorithms for nearest neighbor search based on techniques such as space partitioning with indexing, as well as dimension reduction or sketching [5]. A popular method for point sets in high-dimensional spaces is Locality-Sensitive Hashing (LSH) [6, 3], an approach that offers a *provably* sub-linear query time and sub-quadratic space complexity, and has been shown to achieve good empirical performance in a variety of applications [4]. The method relies on the notion of *locality-sensitive hash functions*. Intuitively, a hash function is locality-sensitive if its probability of collision is higher for "nearby" points than for points that are "far apart". More formally, two points are nearby if their distance is at most $r_1$, and they are far apart if their distance is at least $r_2 = c \cdot r_1$, where $c > 1$ quantifies the gap between "near" and "far". The quality of a hash function is characterized by two key parameters: $p_1$ is the collision probability for nearby points, and $p_2$ is the collision probability for points that are far apart. The gap between $p_1$ and $p_2$ determines how "sensitive" the hash function is to changes in distance, and this property is captured by the parameter $\rho = \frac{\log 1/p_1}{\log 1/p_2}$, which can usually be expressed as a function of the distance gap $c$. The problem of designing good locality-sensitive hash functions and LSH-based efficient nearest neighbor search algorithms has attracted significant attention over the last few years.

---

[*]The authors are listed in alphabetical order.

In this paper, we focus on LSH for the Euclidean distance *on the unit sphere*, which is an important special case for several reasons. First, the spherical case is relevant in practice: Euclidean distance on a sphere corresponds to the *angular distance* or *cosine similarity*, which are commonly used in applications such as comparing image feature vectors [7], speaker representations [8], and tf-idf data sets [9]. Moreover, on the theoretical side, the paper [2] shows a reduction from Nearest Neighbor Search in the *entire* Euclidean space to the spherical case. These connections lead to a natural question: what are good LSH families for this special case?

On the theoretical side, the recent work of [1, 2] gives the best known provable guarantees for LSH-based nearest neighbor search w.r.t. the Euclidean distance on the unit sphere. Specifically, their algorithm has a query time of $O(n^\rho)$ and space complexity of $O(n^{1+\rho})$ for $\rho = \frac{1}{2c^2-1}$.[1] E.g., for the approximation factor $c = 2$, the algorithm achieves a query time of $n^{1/7+o(1)}$. At the heart of the algorithm is an LSH scheme called *Spherical LSH*, which works for unit vectors. Its key property is that it can distinguish between distances $r_1 = \sqrt{2}/c$ and $r_2 = \sqrt{2}$ with probabilities yielding $\rho = \frac{1}{2c^2-1}$ (the formula for the full range of distances is more complex and given in Section 3). Unfortunately, the scheme as described in the paper is not applicable in practice as it is based on rather complex hash functions that are very time consuming to evaluate. E.g., simply evaluating a single hash function from [2] can take more time than a linear scan over $10^6$ points. Since an LSH data structure contains many individual hash functions, using their scheme would be slower than a simple linear scan over all points in $P$ unless the number of points $n$ is extremely large.

On the practical side, the hyperplane LSH introduced in the influential work of Charikar [3] has worse theoretical guarantees, but works well in practice. Since the hyperplane LSH can be implemented very efficiently, it is the standard hash function in practical LSH-based nearest neighbor algorithms[2] and the resulting implementations has been shown to improve over a linear scan on real data by multiple orders of magnitude [14, 9].

The aforementioned discrepancy between the theory and practice of LSH raises an important question: is there a locality-sensitive hash function with *optimal* guarantees that also improves over the hyperplane LSH in practice?

In this paper we show that there is a family of locality-sensitive hash functions that achieves both objectives. Specifically, the hash functions match the theoretical guarantee of Spherical LSH from [2] and, when combined with additional techniques, give better experimental results than the hyperplane LSH. More specifically, our contributions are:

**Theoretical guarantees for the cross-polytope LSH.**   We show that a hash function based on randomly rotated cross-polytopes (i.e., unit balls of the $\ell_1$-norm) achieves the same parameter $\rho$ as the Spherical LSH scheme in [2], assuming data points are unit vectors. While the cross-polytope LSH family has been proposed by researchers before [15, 16] we give the first theoretical analysis of its performance.

**Fine-grained lower bound for cosine similarity LSH.**   To highlight the difficulty of obtaining optimal *and* practical LSH schemes, we prove the first *non-asymptotic* lower bound on the trade-off between the collision probabilities $p_1$ and $p_2$. So far, the optimal LSH upper bound $\rho = \frac{1}{2c^2-1}$ (from [1, 2] and cross-polytope from here) attain this bound only in the limit, as $p_1, p_2 \to 0$. Very small $p_1$ and $p_2$ are undesirable since the hash evaluation time is often proportional to $1/p_2$. Our lower bound proves this is unavoidable: if we require $p_2$ to be large, $\rho$ has to be suboptimal.

This result has two important implications for designing practical hash functions. First, it shows that the trade-offs achieved by the cross-polytope LSH and the scheme of [1, 2] are essentially optimal. Second, the lower bound guides design of future LSH functions: if one is to significantly improve upon the cross-polytope LSH, one has to design a hash function that is computed more efficiently than by explicitly enumerating its range (see Section 4 for a more detailed discussion).

**Multiprobe scheme for the cross-polytope LSH.**   The space complexity of an LSH data structure is sub-*quadratic*, but even this is often too large (i.e., strongly super-*linear* in the number of points),

and several methods have been proposed to address this issue. Empirically, the most efficient scheme is multiprobe LSH [14], which leads to a significantly reduced memory footprint for the hyperplane LSH. In order to make the cross-polytope LSH competitive in practice with the multiprobe hyperplane LSH, we propose a novel multiprobe scheme for the cross-polytope LSH.

We complement these contributions with an experimental evaluation on both real and synthetic data (SIFT vectors, tf-idf data, and a random point set). In order to make the cross-polytope LSH practical, we combine it with fast pseudo-random rotations [17] via the Fast Hadamard Transform, and feature hashing [18] to exploit sparsity of data. Our results show that for data sets with around $10^5$ to $10^8$ points, our multiprobe variant of the cross-polytope LSH is up to $10\times$ faster than an efficient implementation of the hyperplane LSH, and up to $700\times$ faster than a linear scan. To the best of our knowledge, our combination of techniques provides the first "exponent-optimal" algorithm that empirically improves over the hyperplane LSH in terms of query time for an *exact* nearest neighbor search.

## 1.1 Related work

The cross-polytope LSH functions were originally proposed in [15]. However, the analysis in that paper was mostly experimental. Specifically, the probabilities $p_1$ and $p_2$ of the proposed LSH functions were estimated empirically using the Monte Carlo method. Similar hash functions were later proposed in [16]. The latter paper also uses DFT to speed-up the random matrix-vector matrix multiplication operation. Both of the aforementioned papers consider only the *single-probe* algorithm.

There are several works that show lower bounds on the quality of LSH hash functions [19, 10, 20, 11]. However, those papers provide only a lower bound on the $\rho$ parameter for asymptotic values of $p_1$ and $p_2$, as opposed to an actual trade-off between these two quantities. In this paper we provide such a trade-off, with implications as outlined in the introduction.

## 2 Preliminaries

We use $\|.\|$ to denote the Euclidean (a.k.a. $\ell_2$) norm on $\mathbb{R}^d$. We also use $S^{d-1}$ to denote the unit sphere in $\mathbb{R}^d$ centered in the origin. The Gaussian distribution with mean zero and variance of one is denoted by $N(0,1)$. Let $\mu$ be a normalized Haar measure on $S^{d-1}$ (that is, $\mu(S^{d-1}) = 1$). Note that $\mu$ it corresponds to the uniform distribution over $S^{d-1}$. We also let $u \sim S^{d-1}$ be a point sampled from $S^{d-1}$ uniformly at random. For $\eta \in \mathbb{R}$ we denote

$$\Phi_c(\eta) = \Pr_{X \sim N(0,1)}[X \geq \eta] = \frac{1}{\sqrt{2\pi}} \int_\eta^\infty e^{-t^2/2} \, dt.$$

We will be interested in the Near Neighbor Search on the sphere $S^{d-1}$ with respect to the Euclidean distance. Note that the angular distance can be expressed via the Euclidean distance between normalized vectors, so our results apply to the angular distance as well.

**Definition 1.** *Given an $n$-point dataset $P \subset S^{d-1}$ on the sphere, the goal of the $(c,r)$-Approximate Near Neighbor problem (ANN) is to build a data structure that, given a query $q \in S^{d-1}$ with the promise that there exists a datapoint $p \in P$ with $\|p - q\| \leq r$, reports a datapoint $p' \in P$ within distance $cr$ from $q$.*

**Definition 2.** *We say that a hash family $\mathcal{H}$ on the sphere $S^{d-1}$ is $(r_1, r_2, p_1, p_2)$-sensitive, if for every $p, q \in S^{d-1}$ one has $\Pr_{h \sim \mathcal{H}}[h(x) = h(y)] \geq p_1$ if $\|x - y\| \leq r_1$, and $\Pr_{h \sim \mathcal{H}}[h(x) = h(y)] \leq p_2$ if $\|x - y\| \geq r_2$,*

It is known [6] that an *efficient* $(r, cr, p_1, p_2)$-sensitive hash family implies a data structure for $(c,r)$-ANN with space $O(n^{1+\rho}/p_1 + dn)$ and query time $O(d \cdot n^\rho/p_1)$, where $\rho = \frac{\log(1/p_1)}{\log(1/p_2)}$.

## 3 Cross-polytope LSH

In this section, we describe the cross-polytope LSH, analyze it, and show how to make it practical. First, we recall the definition of the cross-polytope LSH [15]: Consider the following hash family

$\mathcal{H}$ for points on a unit sphere $S^{d-1} \subset \mathbb{R}^d$. Let $A \in \mathbb{R}^{d \times d}$ be a random matrix with i.i.d. Gaussian entries ("a random rotation"). To hash a point $x \in S^{d-1}$, we compute $y = Ax/\|Ax\| \in S^{d-1}$ and then find the point closest to $y$ from $\{\pm e_i\}_{1 \leq i \leq d}$, where $e_i$ is the $i$-th standard basis vector of $\mathbb{R}^d$. We use the closest neighbor as a hash of $x$.

The following theorem bounds the collision probability for two points under the above family $\mathcal{H}$.

**Theorem 1.** *Suppose that $p, q \in S^{d-1}$ are such that $\|p - q\| = \tau$, where $0 < \tau < 2$. Then,*

$$\ln \frac{1}{\Pr_{h \sim \mathcal{H}} [h(p) = h(q)]} = \frac{\tau^2}{4 - \tau^2} \cdot \ln d + O_\tau(\ln \ln d) .$$

Before we show how to prove this theorem, we briefly describe its implications. Theorem 1 shows that the cross-polytope LSH achieves essentially the same bounds on the collision probabilities as the (theoretically) optimal LSH for the sphere from [2] (see Section "Spherical LSH" there). In particular, substituting the bounds from Theorem 1 for the cross-polytope LSH into the standard reduction from Near Neighbor Search to LSH [6], we obtain the following data structure with sub-quadratic space and sublinear query time for Near Neighbor Search on a sphere.

**Corollary 1.** *The $(c, r)$-ANN on a unit sphere $S^{d-1}$ can be solved in space $O(n^{1+\rho} + dn)$ and query time $O(d \cdot n^\rho)$, where $\rho = \frac{1}{c^2} \cdot \frac{4 - c^2 r^2}{4 - r^2} + o(1)$ .*

We now outline the proof of Theorem 1. For the full proof, see Appendix B.

Due to the spherical symmetry of Gaussians, we can assume that $p = e_1$ and $q = \alpha e_1 + \beta e_2$, where $\alpha, \beta$ are such that $\alpha^2 + \beta^2 = 1$ and $(\alpha - 1)^2 + \beta^2 = \tau^2$. Then, we expand the collision probability:

$$
\begin{aligned}
\Pr_{h \sim \mathcal{H}} [h(p) = h(q)] &= 2d \cdot \Pr_{h \sim \mathcal{H}} [h(p) = h(q) = e_1] \\
&= 2d \cdot \Pr_{u, v \sim N(0,1)^d} [\forall i \ |u_i| \leq u_1 \text{ and } |\alpha u_i + \beta v_i| \leq \alpha u_1 + \beta v_1] \\
&= 2d \cdot \mathop{\mathrm{E}}_{X_1, Y_1} \left[ \Pr_{X_2, Y_2} \left[ |X_2| \leq X_1 \text{ and } |\alpha X_2 + \beta Y_2| \leq \alpha X_1 + \beta Y_1 \right]^{d-1} \right], \quad (1)
\end{aligned}
$$

where $X_1, Y_1, X_2, Y_2 \sim N(0, 1)$. Indeed, the first step is due to the spherical symmetry of the hash family, the second step follows from the above discussion about replacing a random orthogonal matrix with a Gaussian one and that one can assume w.l.o.g. that $p = e_1$ and $q = \alpha e_1 + \beta e_2$; the last step is due to the independence of the entries of $u$ and $v$.

Thus, proving Theorem 1 reduces to estimating the right-hand side of (1). Note that the probability $\Pr[|X_2| \leq X_1 \text{ and } |\alpha X_2 + \beta Y_2| \leq \alpha X_1 + \beta Y_1]$ is equal to the Gaussian area of the planar set $S_{X_1, Y_1}$ shown in Figure 1a. The latter is *heuristically* equal to $1 - e^{-\Delta^2/2}$, where $\Delta$ is the distance from the origin to the complement of $S_{X_1, Y_1}$, which is easy to compute (see Appendix A for the precise statement of this argument). Using this estimate, we compute (1) by taking the outer expectation.

## 3.1 Making the cross-polytope LSH practical

As described above, the cross-polytope LSH is not quite practical. The main bottleneck is sampling, storing, and applying a random rotation. In particular, to multiply a random Gaussian matrix with a vector, we need time proportional to $d^2$, which is infeasible for large $d$.

**Pseudo-random rotations.** To rectify this issue, we instead use *pseudo-random rotations*. Instead of multiplying an input vector $x$ by a random Gaussian matrix, we apply the following linear transformation: $x \mapsto HD_3 HD_2 HD_1 x$, where $H$ is the Hadamard transform, and $D_i$ for $i \in \{1, 2, 3\}$ is a random diagonal $\pm 1$-matrix. Clearly, this is an orthogonal transformation, which one can store in space $O(d)$ and evaluate in time $O(d \log d)$ using the Fast Hadamard Transform. This is similar to pseudo-random rotations used in the context of LSH [21], dimensionality reduction [17], or compressed sensing [22]. While we are currently not aware how to prove rigorously that such pseudo-random rotations perform as well as the fully random ones, empirical evaluations show that three applications of $HD_i$ are exactly equivalent to applying a true random rotation (when $d$ tends to infinity). We note that only *two* applications of $HD_i$ are not sufficient.

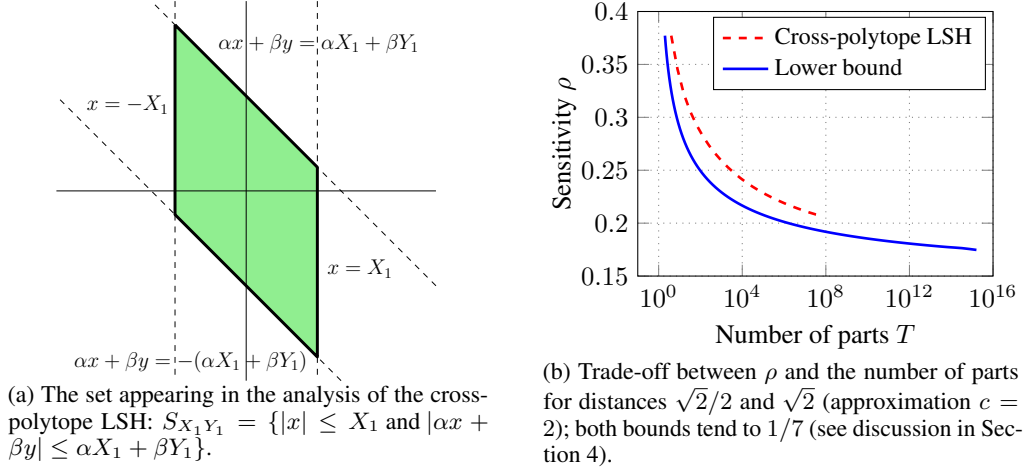

(a) The set appearing in the analysis of the cross-polytope LSH: $S_{X_1 Y_1} = \{|x| \leq X_1 \text{ and } |\alpha x + \beta y| \leq \alpha X_1 + \beta Y_1\}$.

(b) Trade-off between $\rho$ and the number of parts for distances $\sqrt{2}/2$ and $\sqrt{2}$ (approximation $c = 2$); both bounds tend to $1/7$ (see discussion in Section 4).

**Feature hashing.** While we can apply a pseudo-random rotation in time $O(d \log d)$, even this can be too slow. E.g., consider an input vector $x$ that is *sparse*: the number of non-zero entries of $x$ is $s$ much smaller than $d$. In this case, we can evaluate the hyperplane LSH from [3] in time $O(s)$, while computing the cross-polytope LSH (even with pseudo-random rotations) still takes time $O(d \log d)$. To speed-up the cross-polytope LSH for sparse vectors, we apply feature hashing [18]: before performing a pseudo-random rotation, we reduce the dimension from $d$ to $d' \ll d$ by applying a linear map $x \mapsto Sx$, where $S$ is a random sparse $d' \times d$ matrix, whose columns have *one* non-zero $\pm 1$ entry sampled uniformly. This way, the evaluation time becomes $O(s + d' \log d')$. [3]

**"Partial" cross-polytope LSH.** In the above discussion, we defined the cross-polytope LSH as a hash family that returns the closest neighbor among $\{\pm e_i\}_{1 \leq i \leq d}$ as a hash (after a (pseudo-)random rotation). In principle, we do not have to consider all $d$ basis vectors when computing the closest neighbor. By restricting the hash to $d' \leq d$ basis vectors instead, Theorem 1 still holds for the new hash family (with $d$ replaced by $d'$) since the analysis is essentially dimension-free. This slight generalization of the cross-polytope LSH turns out to be useful for experiments (see Section 6). Note that the case $d' = 1$ corresponds to the hyperplane LSH.

## 4  Lower bound

Let $\mathcal{H}$ be a hash family on $S^{d-1}$. For $0 < r_1 < r_2 < 2$ we would like to understand the trade-off between $p_1$ and $p_2$, where $p_1$ is the *smallest* probability of collision under $\mathcal{H}$ for points at distance *at most* $r_1$ and $p_2$ is the *largest* probability of collision for points at distance *at least* $r_2$. We focus on the case $r_2 \approx \sqrt{2}$ because setting $r_2$ to $\sqrt{2} - o(1)$ (as $d$ tends to infinity) allows us to replace $p_2$ with the following quantity that is somewhat easier to handle:

$$p_2^* = \Pr_{\substack{h \sim \mathcal{H} \\ u,v \sim S^{d-1}}} [h(u) = h(v)].$$

This quantity is at most $p_2 + o(1)$, since the distance between two random points on a unit sphere $S^{d-1}$ is tightly concentrated around $\sqrt{2}$. So for a hash family $\mathcal{H}$ on a unit sphere $S^{d-1}$, we would like to understand the upper bound on $p_1$ in terms of $p_2^*$ and $0 < r_1 < \sqrt{2}$.

For $0 \leq \tau \leq \sqrt{2}$ and $\eta \in \mathbb{R}$, we define

$$\Lambda(\tau, \eta) = \Pr_{X,Y \sim N(0,1)} \left[ X \geq \eta \text{ and } \left(1 - \frac{\tau^2}{2}\right) \cdot X + \sqrt{\tau^2 - \frac{\tau^4}{4}} \cdot Y \geq \eta \right] \Big/ \Pr_{X \sim N(0,1)} [X \geq \eta].$$

We are now ready to formulate the main result of this section.

**Theorem 2.** *Let $\mathcal{H}$ be a hash family on $S^{d-1}$ such that every function in $\mathcal{H}$ partitions the sphere into at most $T$ parts of measure at most $1/2$. Then we have $p_1 \leq \Lambda(r_1, \eta) + o(1)$, where $\eta \in \mathbb{R}$ is such that $\Phi_c(\eta) = p_2^*$ and $o(1)$ is a quantity that depends on $T$ and $r_1$ and tends to $0$ as $d$ tends to infinity.*

The idea of the proof is first to reason about one part of the partition using the isoperimetric inequality from [23], and then to apply a certain averaging argument by proving concavity of a function related to $\Lambda$ using a delicate analytic argument. For the full proof, see Appendix C.

We note that the above requirement of all parts induced by $\mathcal{H}$ having measure at most $1/2$ is only a technicality. We conjecture that Theorem 2 holds without this restriction. In any case, as we will see below, in the interesting range of parameters this restriction is essentially irrelevant.

One can observe that if every hash function in $\mathcal{H}$ partitions the sphere into at most $T$ parts, then $p_2^* \geq \frac{1}{T}$ (indeed, $p_2^*$ is precisely the average sum of squares of measures of the parts). This observation, combined with Theorem 2, leads to the following interesting consequence. Specifically, we can numerically estimate $\Lambda$ in order to give a lower bound on $\rho = \frac{\log(1/p_1)}{\log(1/p_2)}$ for any hash family $\mathcal{H}$ in which every function induces at most $T$ parts of measure at most $1/2$. See Figure 1b, where we plot this lower bound for $r_1 = \sqrt{2}/2$,[4] together with an upper bound that is given by the cross-polytope LSH[5] (for which we use numerical estimates for (1)). We can make several conclusions from this plot. First, the cross-polytope LSH gives an almost optimal trade-off between $\rho$ and $T$. Given that the evaluation time for the cross-polytope LSH is $O(T \log T)$ (if one uses pseudo-random rotations), we conclude that in order to improve upon the cross-polytope LSH substantially in practice, one should design an LSH family with $\rho$ being close to optimal and evaluation time that is *sublinear in $T$*. We note that none of the known LSH families for a sphere has been shown to have this property. This direction looks especially interesting since the convergence of $\rho$ to the optimal value (as $T$ tends to infinity) is extremely slow (for instance, according to Figure 1b, for $r_1 = \sqrt{2}/2$ and $r_2 \approx \sqrt{2}$ we need more than $10^5$ parts to achieve $\rho \leq 0.2$, whereas the optimal $\rho$ is $1/7 \approx 0.143$).

## 5    Multiprobe LSH for the cross-polytope LSH

We now describe our multiprobe scheme for the cross-polytope LSH, which is a method for reducing the number of independent hash tables in an LSH data structure. Given a query point $q$, a "standard" LSH data structure considers only a *single* cell in each of the $L$ hash tables (the cell is given by the hash value $h_i(q)$ for $i \in [L]$). In multiprobe LSH, we consider candidates from *multiple* cells in each table [14]. The rationale is the following: points $p$ that are close to $q$ but fail to collide with $q$ under hash function $h_i$ are still likely to hash to a value that is close to $h_i(q)$. By probing multiple hash locations close to $h_i(q)$ in the same table, multiprobe LSH achieves a given probability of success with a smaller number of hash tables than "standard" LSH. Multiprobe LSH has been shown to perform well in practice [14, 24].

The main ingredient in multiprobe LSH is a probing scheme for generating and ranking possible modifications of the hash value $h_i(q)$. The probing scheme should be computationally efficient and ensure that more likely hash locations are probed first. For a single cross-polytope hash, the order of alternative hash values is straightforward: let $x$ be the (pseudo-)randomly rotated version of query point $q$. Recall that the "main" hash value is $h_i(q) = \arg\max_{j \in [d]} |x_j|$.[6] Then it is easy to see that the second highest probability of collision is achieved for the hash value corresponding to the coordinate with the second largest absolute value, etc. Therefore, we consider the indices $i \in [d]$ sorted by their absolute value as our probing sequence or "ranking" for a single cross-polytope.

The remaining question is how to combine multiple cross-polytope rankings when we have more than one hash function. As in the analysis of the cross-polytope LSH (see Section 3, we consider two points $q = e_1$ and $p = \alpha e_1 + \beta e_2$ at distance $R$. Let $A^{(i)}$ be the i.i.d. Gaussian matrix of hash

function $h_i$, and let $x^{(i)} = A^{(i)}e_1$ be the randomly rotated version of point $q$. Given $x^{(i)}$, we are interested in the probability of $p$ hashing to a certain combination of the individual cross-polytope rankings. More formally, let $r_{v_i}^{(i)}$ be the index of the $v_i$-th largest element of $|x^{(i)}|$, where $v \in [d]^k$ specifies the alternative probing location. Then we would like to compute

$$\Pr_{A^{(1)},\dots,A^{(k)}}\left[h_i(p) = r_{v_i}^{(i)} \text{ for all } i \in [k] \mid A^{(i)}q = x^{(i)}\right]$$

$$= \prod_{i=1}^{k} \Pr_{A^{(i)}}\left[\arg\max_{j \in [d]} \left|(\alpha \cdot A^{(i)}e_1 + \beta \cdot A^{(i)}e_2)_j\right| = r_{v_i}^{(i)} \;\middle|\; A^{(i)}e_1 = x^{(i)}\right].$$

If we knew this probability for all $v \in [d]^k$, we could sort the probing locations by their probability. We now show how to approximate this probability efficiently for a single value of $i$ (and hence drop the superscripts to simplify notation). WLOG, we permute the rows of $A$ so that $r_v = v$ and get

$$\Pr_{A}\left[\arg\max_{j \in [d]} \left|(\alpha x + \beta \cdot Ae_2)_j\right| = v \;\middle|\; Ae_1 = x\right] = \Pr_{y \sim N(0,I_d)}\left[\arg\max_{j \in [d]} \left|(x + \frac{\beta}{\alpha} \cdot y)_j\right| = v\right].$$

The RHS is the Gaussian measure of the set $S = \{y \in \mathbb{R}^d \mid \arg\max_{j \in [d]} |(x + \frac{\beta}{\alpha}y)_j| = v\}$. Similar to the analysis of the cross-polytope LSH, we approximate the measure of $S$ by its distance to the origin. Then the probability of probing location $v$ is proportional to $\exp(-\|y_{x,v}\|^2)$, where $y_{x,v}$ is the shortest vector $y$ such that $\arg\max_j |x+y|_j = v$. Note that the factor $\beta/\alpha$ becomes a proportionality constant, and hence the probing scheme does not require to know the distance $R$. For computational performance and simplicity, we make a further approximation and use $y_{x,v} = (\max_i |x_i| - |x_v|) \cdot e_v$, i.e., we only consider modifying a single coordinate to reach the set $S$.

Once we have estimated the probabilities for each $v_i \in [d]$, we incrementally construct the probing sequence using a binary heap, similar to the approach in [14]. For a probing sequence of length $m$, the resulting algorithm has running time $O(L \cdot d \log d + m \log m)$. In our experiments, we found that the $O(L \cdot d \log d)$ time taken to sort the probing candidates $v_i$ dominated the running time of the hash function evaluation. In order to circumvent this issue, we use an incremental sorting approach that only sorts the relevant parts of each cross-polytope and gives a running time of $O(L \cdot d + m \log m)$.

## 6   Experiments

We now show that the cross-polytope LSH, combined with our multiprobe extension, leads to an algorithm that is also efficient in practice and improves over the hyperplane LSH on several data sets. The focus of our experiments is the query time for an *exact* nearest neighbor search. Since hyperplane LSH has been compared to other nearest-neighbor algorithms before [8], we limit our attention to the relative speed-up compared with hyperplane hashing.

We evaluate the two hashing schemes on three types of data sets. We use a synthetic data set of randomly generated points because this allows us to vary a single problem parameter while keeping the remaining parameters constant. We also investigate the performance of our algorithm on real data: two tf-idf data sets [25] and a set of SIFT feature vectors [7]. We have chosen these data sets in order to illustrate when the cross-polytope LSH gives large improvements over the hyperplane LSH, and when the improvements are more modest. See Appendix D for a more detailed description of the data sets and our experimental setup (implementation details, CPU, etc.).

In all experiments, we set the algorithm parameters so that the empirical probability of successfully finding the exact nearest neighbor is at least 0.9. Moreover, we set the number of LSH tables $L$ so that the amount of additional memory occupied by the LSH data structure is comparable to the amount of memory necessary for storing the data set. We believe that this is the most interesting regime because significant memory overheads are often impossible for large data sets. In order to determine the parameters that are not fixed by the above constraints, we perform a grid search over the remaining parameter space and report the best combination of parameters. For the cross-polytope hash, we consider "partial" cross-polytopes in the last of the $k$ hash functions in order to get a smooth trade-off between the various parameters (see Section 3.1).

**Multiprobe experiments.**   In order to demonstrate that the multiprobe scheme is critical for making the cross-polytope LSH competitive with hyperplane hashing, we compare the performance of a

| Data set | Method | Query time (ms) | **Speed-up vs HP** | Best $k$ | Number of candidates | Hashing time (ms) | Distances time (ms) |
|---|---|---|---|---|---|---|---|
| NYT | HP | 120 ms | | 19 | 57,200 | 16 | 96 |
| NYT | CP | 35 ms | **3.4×** | 2 (64) | 17,900 | 3.0 | 30 |
| pubmed | HP | 857 ms | | 20 | 1,481,000 | 36 | 762 |
| pubmed | CP | 213 ms | **4.0×** | 2 (512) | 304,000 | 18 | 168 |
| SIFT | HP | 3.7 ms | | 30 | 18,600 | 0.2 | 3.0 |
| SIFT | CP | 3.1 ms | **1.2×** | 6 (1) | 13,400 | 0.6 | 2.2 |

Table 1: Average running times for a single nearest neighbor query with the hyperplane (HP) and cross-polytope (CP) algorithms on three real data sets. The cross-polytope LSH is faster than the hyperplane LSH on all data sets, with significant speed-ups for the two tf-idf data sets NYT and pubmed. For the cross-polytope LSH, the entries for $k$ include both the number of individual hash functions per table and (in parenthesis) the dimension of the last of the $k$ cross-polytopes.

"standard" cross-polytope LSH data structure with our multiprobe variant on an instance of the random data set ($n = 2^{20}$, $d = 128$). As can be seen in Table 2 (Appendix D), the multiprobe variant is about $13×$ faster in our memory-constrained setting ($L = 10$). Note that in all of the following experiments, the speed-up of the multiprobe cross-polytope LSH compared to the multiprobe hyperplane LSH is less than $11×$. Hence without our multiprobe addition, the cross-polytope LSH would be slower than the hyperplane LSH, for which a multiprobe scheme is already known [14].

**Experiments on random data.** Next, we show that the better time complexity of the cross-polytope LSH already applies for moderate values of $n$. In particular, we compare the cross-polytope LSH, combined with fast rotations (Section 3.1) and our multiprobe scheme, to a multi-probe hyper-plane LSH on random data. We keep the dimension $d = 128$ and the distance to the nearest neighbor $R = \sqrt{2}/2$ fixed, and vary the size of the data set from $2^{20}$ to $2^{28}$. The number of hash tables $L$ is set to 10. For $2^{20}$ points, the cross-polytope LSH is already $3.5×$ faster than the hyperplane LSH, and for $n = 2^{28}$ the speedup is $10.3×$ (see Table 3 in Appendix D). Compared to a linear scan, the speed-up achieved by the cross-polytope LSH ranges from $76×$ for $n = 2^{20}$ to about $700×$ for $n = 2^{28}$.

**Experiments on real data.** On the SIFT data set ($n = 10^6$ and $d = 128$), the cross-polytope LSH achieves a modest speed-up of $1.2×$ compared to the hyperplane LSH (see Table 1). On the other hand, the speed-up is is $3 - 4×$ on the two tf-idf data sets, which is a significant improvement considering the relatively small size of the NYT data set ($n \approx 300,000$). One important difference between the data sets is that the typical distance to the nearest neighbor is smaller in the SIFT data set, which can make the nearest neighbor problem easier (see Appendix D). Since the tf-idf data sets are very high-dimensional but sparse ($d \approx 100,000$), we use the feature hashing approach described in Section 3.1 in order to reduce the hashing time of the cross-polytope LSH (the standard hyperplane LSH already runs in time proportional to the sparsity of a vector). We use 1024 and 2048 as feature hashing dimensions for NYT and pubmed, respectively.

## Acknowledgments

We thank Michael Kapralov for many valuable discussions during various stages of this work. We also thank Stefanie Jegelka and Rasmus Pagh for helpful conversations. This work was supported in part by the NSF and the Simons Foundation. Work done in part while the first author was at the Simons Institute for the Theory of Computing.

## Footnotes

[1]This running time is known to be essentially optimal for a large class of algorithms [10, 11].

[2]Note that if the data points are *binary*, more efficient LSH schemes exist [12, 13]. However, in this paper we consider algorithms for general (non-binary) vectors.

[3] Note that one can apply Lemma 2 from the arXiv version of [18] to claim that—after such a dimension reduction—the distance between *any* two points remains sufficiently concentrated for the bounds from Theorem 1 to still hold (with $d$ replaced by $d'$).

[4]The situation is qualitatively similar for other values of $r_1$.

[5]More specifically, for the "partial" version from Section 3.1, since $T$ should be constant, while $d$ grows

[6]In order to simplify notation, we consider a slightly modified version of the cross-polytope LSH that maps both the standard basis vector $+e_j$ and its opposite $-e_j$ to the same hash value. It is easy to extend the multiprobe scheme defined here to the "full" cross-polytope LSH from Section 3.

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
