[Supplementary Material]

## A  Gaussian measure of a planar set

In this Section we formalize the intuition that the standard Gaussian measure of a closed subset $A \subseteq \mathbb{R}^2$ behaves like $e^{-\Delta_A^2/2}$, where $\Delta_A$ is the distance from the origin to $A$, unless $A$ is quite special.

For a closed subset $A \subseteq \mathbb{R}^2$ and $r > 0$ denote $0 \leq \mu_A(r) \leq 1$ the normalized measure of the intersection $A \cap rS^1$ ($A$ with the circle centered in the origin and of radius $r$):

$$\mu_A(r) := \frac{\mu(A \cap rS^1)}{2\pi r};$$

here $\mu$ is the standard one-dimensional Lebesgue measure (see Figure 2a). Denote $\Delta_A := \inf\{r > 0 : \mu_A(r) > 0\}$ the (essential) distance from the origin to $A$. Let $\mathcal{G}(A)$ be the standard Gaussian measure of $A$.

**Lemma 1.** *Suppose that $A \subseteq \mathbb{R}^2$ is a closed set such that $\mu_A(r)$ is non-decreasing. Then,*

$$\sup_{r>0}\left(\mu_A(r) \cdot e^{-r^2/2}\right) \leq \mathcal{G}(A) \leq e^{-\Delta_A^2/2}.$$

*Proof.* For the upper bound, we note that

$$\mathcal{G}(A) = \int_0^\infty \mu_A(r) \cdot re^{-r^2/2}\, dr \leq \int_{\Delta_A}^\infty re^{-r^2/2}\, dr = e^{-\Delta_A^2/2}.$$

For the lower bound, we similarly have, for every $r^* > 0$,

$$\mathcal{G}(A) = \int_0^\infty \mu_A(r) \cdot re^{-r^2/2}\, dr \geq \mu_A(r^*) \cdot \int_{r^*}^\infty re^{-r^2/2}\, dr = \mu_A(r^*)e^{-(r^*)^2/2},$$

where we use that $\mu_A(r^*)$ is non-decreasing. $\qquad\square$

Now we derive two corollaries of Lemma 1.

**Lemma 2.** *Let $K \subseteq \mathbb{R}^2$ be the complement of an open convex subset of the plane that is symmetric around the origin. Then, for every $0 < \varepsilon < 1/3$,*

$$\Omega\left(\varepsilon^{1/2} \cdot e^{-(1+\varepsilon)\cdot\Delta_K^2/2}\right) \leq \mathcal{G}(K) \leq e^{-\Delta_K^2/2}.$$

*Proof.* This follows from Lemma 1: indeed, due to the convexity of the complement of $K$, $\mu_K(r)$ is non-decreasing. It is easy to check that

$$\mu_K\left((1+\varepsilon)\Delta_K\right) = \Omega\left(\varepsilon^{1/2}\right),$$

again, due to the convexity (see Figure 2b). Thus, the required bounds follow. $\qquad\square$

**Lemma 3.** *Let $K \subseteq \mathbb{R}^2$ be an intersection of two closed half-planes such that:*

- *$K$ does not contain a line;*

- *the "corner" of $K$ is the closest point of $K$ to the origin;*

- *the angle between half-planes equals to $0 < \alpha < \pi$.*

*Then, for every $0 < \varepsilon < 1/2$,*

$$\Omega_\alpha\left(\varepsilon \cdot e^{-(1+\varepsilon)\cdot\Delta_K^2}\right) \leq \mathcal{G}(K) \leq e^{-\Delta_K^2/2}.$$

*Proof.* This, again, follows from Lemma 1. The second condition implies that $\mu_K(r)$ is non-decreasing, and an easy computation shows that

$$\mu_K((1+\varepsilon)\Delta_K) \geq \Omega_\alpha(\varepsilon)$$

(see Figure 2c). $\qquad\square$

Figure 2

(a) Definition of $\mu_A(r)$

(b) For Lemma 2

(c) For Lemma 3

# B   Proof of Theorem 1

In this section we complete the proof of Theorem 1, following the outline from Section 3. Our starting point is the collision probability bound from Eqn. (1).

For $u, v \in \mathbb{R}$ with $u \geq 0$ and $\alpha u + \beta v \geq 0$ define,

$$\sigma(u,v) = \Pr_{X_2, Y_2 \sim N(0,1)} [|X_2| \leq u \text{ and } |\alpha X_2 + \beta Y_2| \leq \alpha u + \beta v].$$

Then, the right-hand side of (1) is equal to

$$2d \cdot \operatorname*{E}_{X_1, Y_1 \sim N(0,1)} [\sigma(X_1, Y_1)^{d-1}].$$

Let us define

$$\Delta(u,v) = \min\{u, \alpha u + \beta v\}.$$

**Lemma 4.** *For every $0 < \varepsilon < 1/3$,*

$$1 - e^{-\Delta(u,v)^2/2} \leq \sigma(u,v) \leq 1 - \Omega\left(\varepsilon^{1/2} \cdot e^{-(1+\varepsilon)\Delta(u,v)^2/2}\right).$$

*Proof.* This is a combination of Lemma 2 together with the following obvious observation: the distance from the origin to the set $\{(x,y) : |x| \geq u \text{ or } |\alpha x + \beta y| \geq \alpha u + \beta v\}$ is equal to $\Delta(u,v)$ (see Figure 1a). $\square$

**Lemma 5.** *For every $t \geq 0$ and $0 < \varepsilon < 1/3$,*

$$\Omega_\tau \left( \varepsilon \cdot e^{-(1+\varepsilon) \cdot \frac{4}{4-\tau^2} \cdot \frac{t^2}{2}} \right) \leq \Pr_{X_1, Y_1 \sim N(0,1)}[\Delta(X_1, Y_1) \geq t] \leq e^{-\frac{4}{4-\tau^2} \cdot \frac{t^2}{2}}.$$

*Proof.* Similar to the previous lemma, this is a consequence of Lemma 3 together with the fact that the squared distance from the origin to the set $\{(x,y) : x \geq t \text{ and } \alpha x + \beta y \geq t\}$ is equal to $\frac{4}{4-\tau^2} \cdot t^2$. $\square$

### B.1 Idealized proof

Let us expand Eqn. (1) further, assuming that the "idealized" versions of Lemma 4 and Lemma 5 hold. Namely, we assume that

$$\sigma(u,v) = 1 - e^{-\Delta(u,v)^2/2}; \tag{2}$$

and

$$\Pr_{X_1, Y_1 \sim N(0,1)}[\Delta(X_1, Y_1) \geq t] = e^{-\frac{4}{4-\tau^2} \cdot \frac{t^2}{2}}. \tag{3}$$

In the next section we redo the computations using the precise bounds for $\sigma(u,v)$ and $\Pr[\Delta(X_1, Y_1) \geq t]$.

Expanding Eqn. (1), we have

$$
\begin{aligned}
\mathop{\mathrm{E}}_{X_1, Y_1 \sim N(0,1)}[\sigma(X_1, Y_1)^{d-1}] &= \int_0^1 \Pr_{X_1, Y_1 \sim N(0,1)}[\sigma(X_1, Y_1) \geq t^{\frac{1}{d-1}}]\, dt \\
&= \int_0^1 \Pr_{X_1, Y_1 \sim N(0,1)}[e^{-\Delta(X_1, Y_1)^2/2} \leq 1 - t^{\frac{1}{d-1}}]\, dt \\
&= \int_0^1 (1 - t^{\frac{1}{d-1}})^{\frac{4}{4-\tau^2}}\, dt \\
&= (d-1) \cdot \int_0^1 (1-u)^{\frac{4}{4-\tau^2}} u^{d-2}\, dt \\
&= (d-1) \cdot B\left(\frac{8-\tau^2}{4-\tau^2}; d-1\right) \\
&= \Theta_\tau(1) \cdot d^{-\frac{4}{4-\tau^2}}, \tag{4}
\end{aligned}
$$

where:

- the first step is a standard expansion of an expectation;
- the second step is due to (2);
- the third step is due to (3);
- the fourth step is a change of variables;
- the fifth step is a definition of the Beta function;
- the sixth step is due to the Stirling approximation.

Overall, substituting (4) into (1), we get:

$$\ln \frac{1}{\Pr_{h \sim \mathcal{H}}[h(p) = h(q)]} = \frac{\tau^2}{4-\tau^2} \cdot \ln d \pm O_\tau(1).$$

### B.2 The real proof

We now perform the exact calculations, using the bounds (involving $\varepsilon$) from Lemma 4 and Lemma 5. We set $\varepsilon = 1/d$ and obtain the following asymptotic statements:

$$\sigma(u,v) = 1 - d^{\pm O(1)} \cdot e^{-(1 \pm d^{-\Omega(1)}) \cdot \Delta(u,v)^2/2};$$

and

$$\Pr_{X,Y \sim N(0,1)}[\Delta(X,Y) \geq t] = d^{\pm O(1)} \cdot e^{-(1 \pm d^{-\Omega(1)}) \cdot \frac{4}{4-\tau^2} \cdot \frac{t^2}{2}}.$$

Then, we can repeat the "idealized" proof (see Eqn. (4)) verbatim with the new estimates and obtain the final form of Theorem 1:

$$\ln \frac{1}{\Pr_{h \sim \mathcal{H}}[h(p) = h(q)]} = \frac{\tau^2}{4-\tau^2} \cdot \ln d \pm O_\tau(\ln \ln d).$$

Note the difference in the low order term between idealized and the real version. As we argue in Section 4, the latter $O_\tau(\ln \ln d)$ is, in fact, tight.

## C  Proof of Theorem 2

**Lemma 6.** *Let $A \subset S^{d-1}$ be a measurable subset of a sphere with $\mu(A) = \mu_0 \leq 1/2$. Then, for $0 < \tau < \sqrt{2}$, one has*

$$\Pr_{u,v \sim S^{d-1}}\left[v \in A \mid u \in A, \|u - v\| \leq \tau\right] = \frac{\Pr_{X,Y \sim N(0,1)}[X \geq \eta \text{ and } \alpha X + \beta Y \geq \eta] + o(1)}{\Pr_{X \sim N(0,1)}[X \geq \eta] + o(1)}, \quad (5)$$

*where:*

- $\alpha = 1 - \frac{\tau^2}{2}$;

- $\beta = \sqrt{\tau^2 - \frac{\tau^4}{4}}$;

- $\eta \in \mathbb{R}$ *is such that* $\Pr_{X \sim N(0,1)}[X \geq \eta] = \mu_0$.

*In particular, if $\mu_0 = \Omega(1)$, then*

$$\Pr_{u,v \sim S^{d-1}}\left[v \in A \mid u \in A, \|u - v\| \leq \tau\right] = \Lambda(\tau, \Phi_c^{-1}(\mu_0)) + o(1).$$

*Proof.* First, the left-hand side of (5) is maximized by a spherical cap of measure $\mu_0$. This follows from Theorem 5 of [23]. So, from now on we assume that $A$ is a spherical cap.

Second, one has

$$\Pr_{u,v \sim S^{d-1}}\left[v \in A \mid u \in A, \|u - v\| \leq \tau\right]$$

$$= \Pr_{u,v \sim S^{d-1}}\left[v \in A \mid u \in A, \|u - v\| = \tau \pm o(1)\right] + o(1)$$

$$= \frac{\Pr_{u \sim S^{d-1}}\left[u_1 \geq \widetilde{\eta} \text{ and } (\alpha \pm o(1))u_1 + (\beta \pm o(1))u_2 \geq \widetilde{\eta}\right]}{\Pr_{u \sim S^{d-1}}[u_1 \geq \widetilde{\eta}]} + o(1)$$

$$= \frac{\Pr_{X,Y \sim N(0,1)}[X \geq \eta \text{ and } \alpha X + \beta Y \geq \eta] + o(1)}{\Pr_{X \sim N(0,1)}[X \geq \eta] + o(1)},$$

where $\widetilde{\eta}$ is such that $\Pr_{u \sim S^{d-1}}[u_1 \geq \widetilde{\eta}] = \mu_0$ and:

- the first step is due to the concentration of measure on the sphere;

- the second step is expansion of the conditional probability;

- the third step is due to the fact that a $O(1)$-dimensional projection of the uniform measure on a sphere of radius $\sqrt{d}$ in $\mathbb{R}^d$ converges in total variation to a standard Gaussian measure [26].

$\square$

**Lemma 7.** *For every $0 < \tau < \sqrt{2}$, the function $\mu \mapsto \Lambda(\tau, \Phi_c^{-1}(\mu))$ is concave for $0 < \mu < 1/2$.*

*Proof.* Abusing notation, for this proof we denote $\Lambda(\eta) = \Lambda(\tau, \eta)$ and
$$I(\eta) = \Pr_{X,Y \sim N(0,1)}[X \geq \eta \text{ and } \alpha X + \beta Y \geq \eta]$$

(that is, $\Lambda(\eta) = I(\eta)/\Phi_c(\eta)$). One has $\Phi_c'(\eta) = -\frac{e^{-\eta^2/2}}{\sqrt{2\pi}}$ and
$$I'(\eta) = -\sqrt{\frac{2}{\pi}} \cdot e^{-\eta^2/2} \cdot \Phi_c\left(\frac{(1-\alpha)\eta}{\beta}\right).$$

Combining, we get
$$\Lambda'(\eta) = \frac{e^{-\eta^2/2}}{\sqrt{2\pi}} \cdot \frac{I(\eta) - 2\Phi_c(\eta)\Phi_c\left(\frac{(1-\alpha)\eta}{\beta}\right)}{\Phi_c(\eta)^2}$$

and
$$\frac{d\Lambda(\Phi_c^{-1}(\mu))}{d\mu} = \frac{2\Phi_c(\eta^*)\Phi_c\left(\frac{(1-\alpha)\eta^*}{\beta}\right) - I(\eta^*)}{\Phi_c(\eta^*)^2} =: \Pi(\eta^*),$$

where $\eta^* = \eta^*(\mu) = \Phi_c^{-1}(\mu)$. It is sufficient to show that $\Pi(\eta^*)$ is non-decreasing in $\eta^*$ for $\eta^* \geq 0$.

We have
$$\Pi'(\eta) = \sqrt{\frac{2}{\pi}} \cdot \frac{e^{-\eta^2/2}}{\Phi_c(\eta)^3} \left(2 \cdot \Phi_c(\eta)\Phi_c\left(\frac{(1-\alpha)\eta}{\beta}\right) - I(\eta) - \frac{1-\alpha}{\beta} \cdot e^{\frac{\alpha(1-\alpha)}{\beta^2}\cdot\eta^2} \Phi_c(\eta)^2\right)$$

$$=: \sqrt{\frac{2}{\pi}} \cdot \frac{e^{-\eta^2/2}}{\Phi_c(\eta)^3} \cdot \Omega(\eta).$$

We need to show that $\Omega(\eta) \geq 0$ for $\eta \geq 0$. We will do this by showing that $\Omega'(\eta) \leq 0$ for $\eta \geq 0$ and that $\lim_{\eta \to \infty} \Omega(\eta) = 0$. The latter is obvious, so let us show the former.
$$\Omega'(\eta) = -\frac{2\alpha(1-\alpha)^2}{\beta^3} \cdot e^{\frac{\alpha(1-\alpha)}{\beta^2}\eta^2} \cdot \Phi_c(\eta)^2 \cdot \eta \leq 0$$

for $\eta \geq 0$. $\square$

Now we are ready to prove Theorem 2. Let us first assume that all the parts have measure $\Omega(1)$. Later we will show that this assumption can be removed. W.l.o.g. we assume that functions from the family have subsets integers as a range. We have,
$$p_1 \leq \Pr_{\substack{u,v \sim S^{d-1} \\ h \sim \mathcal{H}}}\left[h(u) = h(v) \mid \|u - v\| \leq \tau\right]$$

$$= \mathop{E}_{h \sim \mathcal{H}}\left[\sum_i \mu(h^{-1}(i))\Pr[v \in h^{-1}(i) \mid u \in h^{-1}(i), \|u - v\| \leq \tau]\right]$$

$$\leq \mathop{E}_{h \sim \mathcal{H}}\left[\sum_i \mu(h^{-1}(i))\Lambda(\tau, \Phi_c^{-1}(\mu(h^{-1}(i))))\right] + o(1)$$

$$\leq \Lambda\left(\tau, \Phi_c^{-1}\left(\mathop{E}_{h \sim \mathcal{H}}\left[\sum_i \mu(h^{-1}(i))^2\right]\right)\right) + o(1)$$

$$\leq \Lambda(\tau, \Phi_c^{-1}(p^*(\mathcal{H}))) + o(1),$$

where:

- the first step is by the definition of $p_1$;
- the third step is due to the condition $\mu(h^{-1}(i)) = \Omega(1)$ and Lemma 6;
- the fourth step is due to Lemma 7 and the assumption $\mu(h^{-1}(i)) \leq 1/2$;
- the final step is due to the definition of $p^*(\mathcal{H})$.

To get rid of the assumption that a measure of every part is $\Omega(1)$ observe that all parts with measure at most $\varepsilon$ contribute to the expectation at most $\varepsilon \cdot T$, since there are at most $T$ pieces in total. Note that if $\varepsilon = o(1)$, then $\varepsilon \cdot T = o(1)$, since we assume $T$ being fixed.

## D   Further description of experiments

In order to compare meaningful running time numbers, we have written fast C++ implementations of both the cross-polytope LSH and the hyperplane LSH. This enables a fair comparison since both implementations have been optimized by us to the same degree. In particular, hyperplane hashing can be implemented efficiently using a matrix-vector multiplication sub-routine for which we use the eigen library (eigen is also used for all other linear algebra operations). For the fast pseudo-random rotation in the cross-polytope LSH, we have written a SIMD-optimized version of the Fast Hadamard Transform (FHT). We compiled our code with g++ 4.9 and the -O3 flag. All experiments except those in Table 3 ran on an Intel Core i5-2500 CPU (3.3 - 3.7 GHz, 6 MB cache) with 8 GB of RAM. Since 8 GB of RAM was too small for the larger values of $n$, we ran the experiments in Table 3 on a machine with an Intel Xeon E5-2690 v2 CPU (3.0 GHz, 25 MB cache) and 512 GB of RAM.

Figure 3: Distance to the nearest neighbor for the four data sets used in our experiments. The SIFT data set has the closest nearest neighbors.

In our experiments, we evaluate the performance of the cross-polytope LSH on the following data sets. Figure 3 shows the distribution of distances to the nearest neighbor for the four data sets.

**random**  For the random data sets, we generate a set of $n$ points uniformly at random on the unit sphere. In order to generate a query, we pick a random point $q'$ from the data set and generate

a point at distance $R$ from $q'$ on the unit sphere. In our experiments, we vary the dimension of the point set between 128 and 1,024. Experiments with the random data set are useful because we can study the impact of various parameters (e.g., the dimension $d$ or the number of points $n$) while keeping the remaining parameters constant.

**pubmed / NYT** The pubmed and NYT data sets contain bag-of-words representations of medical paper abstracts and newspaper articles, respectively [25]. We convert this representation into standard tf-idf feature vectors with dimensionality about 100,000. The number of points in the pubmed data set is about 8 million, for NYT it is 300,000. Before setting up the LSH data structures, we set 1000 data points aside as query vectors. When selecting query vectors, we limit our attention to points for which the inner product with the nearest neighbor is between 0.3 and 0.8. We believe that this is the most interesting range since near-duplicates (inner product close to 1) can be identified more efficiently with other methods, and points without a close nearest neighbor (inner product less than 0.3) often do not have a semantically meaningful match.

**SIFT** We use the standard data set of one million SIFT feature vectors from [7], which also contains a set of 10,000 query vectors. The SIFT feature vectors have dimension 128 and (approximately) live on a sphere. We normalize the feature vectors to unit length but keep the original nearest neighbor assignments—this is possible because only a very small fraction of nearest neighbors changes through normalization. We include this data set as an example where the speed-up of the cross-polytope LSH is more modest.

| Method | $k$ | Last CP dimension | Extra probes | **Query time (ms)** | Number of candidates | CP hashing time (ms) | Distances time (ms) |
|---|---|---|---|---|---|---|---|
| Single-probe | 1 | 128 | 0 | **6.7** | 39800 | 0.01 | 6.3 |
| Multiprobe | 3 | 16 | 896 | **0.51** | 867 | 0.22 | 0.16 |

Table 2: Comparison of "standard" LSH using the cross-polytope (CP) hash vs. our multiprobe variant ($L = 10$ in both cases). On a random data set with $n = 2^{20}$, $d = 128$, and $R = \sqrt{2}/2$, the single-probe scheme requires $13\times$ more time per query. Due to the larger value of $k$, the multiprobe variant performs fewer distance computations, which leads to a better trade-off between the hash computation time and the time spent on computing distances to candidates from the hash tables.

| Data set size $n$ | $2^{20}$ | $2^{22}$ | $2^{24}$ | $2^{26}$ | $2^{28}$ |
|---|---|---|---|---|---|
| HP query time (ms) | 2.6 | 7.4 | 25 | 63 | 185 |
| CP query time (ms) | 0.75 | 1.4 | 3.1 | 8.8 | 18 |
| **Speed-up** | **3.5×** | **5.3×** | **8.1×** | **7.2×** | **10.3×** |
| $k$ for CP | 3 (16) | 3 (64) | 3 (128) | 4 (2) | 4 (64) |

Table 3: Average running times for a single nearest neighbor query with the hyperplane (HP) and cross-polytope (CP) algorithms on a random data set with $d = 128$ and $R = \sqrt{2}/2$. The cross-polytope LSH is up to $10\times$ faster than the hyperplane LSH. The last row of the table indicates the optimal choice of $k$ for the cross-polytope LSH and (in parenthesis) the dimension of the last of the $k$ cross-polytopes; all other cross-polytopes have full dimension 128. Note that the speed-up ratio is not monotonically increasing because the cross-polytope LSH performs better for values of $n$ where the optimal setting of $k$ uses a last cross-polytope with high dimension.