[Reviews · NeurIPS 2015]

Submitted by Assigned_Reviewer_1

The precision/recall results for these LSH schemes are missing. Does both methods find the same neighbors? Some discussion on P/R will be useful. There are no comparisons against data dependent methods, hence it is unclear if this method will be preferred over data-dependent hashing schemes. Discussion of proposed method with respect to data dependent methods will be handy for the applicability of proposed approach in other nearest neighbor applications.
Summary: The paper presents a multiprobe LSH variant of Cross-polytope LSH. Also presents a lower bound for the quality of any LSH family for angular distances. The results show that proposed method is faster than vanilla LSH.

Submitted by Assigned_Reviewer_2

The authors propose a novel LSH family for angular distance which (a) matches the theoretical guarantees of Spherical LSH (i.e., an asymptotically optimal runtime exponent) while at the same time (unlike Spherical LSH) being practical in that they outperform Hyperplane LSH for the same task by up to an order of magnitude.

This result is achieved by the combination of techniques where (a) a data point is first hashed using a random rotation and then the after-image matched to the closest point y from {+/-e_i}, where e_i is the ith standard basis vector of \R^d, (b) the random rotation is performed as a sequence of 3 pseudo-random rotations (reducing

the asymototic overhead from O(d^2) to O(d log d), (c) feature hashing to reduce the effective dimensionality d, and (d) using a variant of multi-probe LSH for retrieval which generates additional probing locations.

The proposed work is novel, original and has the rare property that it combines theoretical optimality with a practical, realistic (C++ instead of MATLAB) implementation.
Summary: The paper proposes an interesting extension on the existing state of the art in LSH for angular distance. It combines theoretical optimality with a practical implementation that is faster than the (practical) state of the art by an order of magnitude.

Submitted by Assigned_Reviewer_3

I think this is an impressive paper, very well-written (even if it is still difficult for a non-expert). One thing about the experiments is ambiguous:

Are all the reported results for multiprobe CP against multiprobe HP?

It seems it is the case from the "multiprobe experiments" paragraph of Section 6 but this should be stated more clearly.

Line 428: what is "input-sparsity time"?

line 176: "outline the proof" (no "of")

line 302, "hash" (!): has

line 342, "considers": consider
Summary: A dense but well-written paper on an algorithm validated with several proofs and experiments for speeding up approximated nearest neighbor search on normalized vectors - which is a common problem in computer vision.

Submitted by Assigned_Reviewer_4

The authors focus on both optimal and practical properties for LSH schemes which are hard to obtain. They show this difficulty by proving a lower bound for LSH on angular distance. The authors also propose the multiprobe LSH which is a variant of cross-polytope LSH to achieve the goal. The experimental results show that the proposed algorithm is faster than other LSH schemes.
Summary: The idea looks sound. They have nice experimental results.

Author Feedback
Author rebuttal: We thank the reviewers for their valuable comments and address them point-by-point below.

Reviewer_1:
* We thank the reviewer for the constructive comments.

* In our experiments, we measured the average query time of nearest neighbor algorithms under the condition that the success probability of finding the exact nearest neighbor is at least 90%. This is a common setup in experimental evaluations of nearest neighbor algorithms (see, e.g., reference [9]). The precision / recall curves suggested by the reviewer are also a common and useful way to evaluate hashing schemes. However, they do not take into account certain important factors such as the time needed to compute the hash function. Due to the space restrictions of the submission, we have therefore decided not to include those plots and instead present results for a wider collection of data sets.

Nevertheless, for the purpose of this rebuttal, we are including several plots that depict the relationship between the probability of finding the true nearest neighbor (i.e., the "recall") and the length of the probing sequences for the HP and CP hashing schemes (the length of the probing sequence corresponds to the number of buckets examined). The plots are for the NYT data set and for the random data set (n = 2^20) and were generated with the same LSH parameters as described in our paper:

https://storage.googleapis.com/nips2015_tmp/nyt_multiprobe_pr2_3.pdf
https://storage.googleapis.com/nips2015_tmp/random_multiprobe_pr2_1.pdf

It is worth noting that for multiple hash tables (L > 1), the lower left corner of the plot (small success probabilities) is most relevant:

https://storage.googleapis.com/nips2015_tmp/nyt_multiprobe_pr2_3_zoomed.pdf
https://storage.googleapis.com/nips2015_tmp/random_multiprobe_pr2_1_zoomed.pdf

The plots demonstrate that the CP hash indeed offers significantly better tradeoffs than the HP hash.

* Yes, in the event of a successful query, both methods find the same neighbors because we require them to find the *exact* nearest neighbor.

* There is indeed is a large body of work on data-dependent hashing methods such as spectral hashing. In our submission, we focus instead on data-oblivious methods, which have several desirable properties compared to the data-dependent techniques:
1. It is easier to set up the data structure in a distributed environment because points can be hashed without performing any global optimization or even communication (see, e.g., reference [9]).
2. Data-oblivious algorithms immediately support insertions and deletions of points, without solving a new optimization problem.
3. In a data-oblivious scheme like ours, the collision probability of two data points depends only on the distance between these two points and not the entire data set. This can lead to more predictable empirical performance and provable guarantees.

For further discussion about the pros and cons of the data-dependent methods, see "Frontiers in Massive Data Analysis", National Academies Press, 2013, page 49.

We believe these and other benefits justify the study of data-oblivious hashing methods on its own.

Reviewer_2:
* We would like to thank the reviewer for the positive feedback.

* Yes, all comparisons between the HP and CP hashing schemes (Tables 1 and 2) are for the multiprobe variants of HP and CP. We mention this in lines 413 - 415 of the paper, but we will also make this point more clear in the final version.

* Line 428: Here, "input-sparsity time" refers to a time complexity that is linear in the number of non-zero elements in a vector (as opposed to, e.g., a linear dependence on the ambient dimension).

* Thank you for pointing out the typos in lines 176, 302, and 342, which we will fix in the final version of the paper.

Reviewer_4:
* We thank the reviewer for the positive comments, in particular the last sentence regarding theoretical optimality and practical experiments - achieving this combination was indeed the main goal for this paper.

Reviewer_5:
Thank you for the comment. We respectfully disagree with the remark that the topic of our paper is not a good fit for NIPS. Over the past few years, several papers about LSH and similar hashing schemes have appeared at NIPS. Moreover, these papers often considered the running time required for nearest neighbor searches. This list of papers includes, but is not limited to, the following works:

NIPS 2014: Shrivastava and Li (received the best paper award);
NIPS 2013: Neyshabur, Yadollahpour, Makarychev, Salakhutdinov, and Srebro
NIPS 2012: Ji, Li, Yan, Zhang, and Tian
NIPS 2012: Gong, Kumar, Verma, and Lazebnik
NIPS 2012: Kong and Li
NIPS 2011: Li, Shrivastava, Moore, and Konig

Reviewer_6:
We thank the reviewer for the positive comments.